# Food and Alcohol Disturbance in High School Adolescents: Prevalence, Characteristics and Association with Problem Drinking and Eating Disorders

**DOI:** 10.3390/ijerph21010083

**Published:** 2024-01-11

**Authors:** Federica Pinna, Federico Suprani, Pasquale Paribello, Paola Milia, Lucia Sanna, Mirko Manchia, Graziella Boi, Annadele Pes, Lorena Lai, Valeria Deiana, Silvia Lostia di Santa Sofia, Laura Puddu, Francesca Fatteri, Alice Ghiani, Alice Lai, Bernardo Carpiniello

**Affiliations:** 1Section of Psychiatry, Department of Medical Sciences and Public Health, University of Cagliari, 09127 Cagliari, Italy; fede.pinna73@gmail.com (F.P.); pasqualeparibello@gmail.com (P.P.); paolamilia.82@gmail.com (P.M.); doc_luciasanna@hotmail.com (L.S.); mirkomanchia@unica.it (M.M.); lorenalai78@yahoo.it (L.L.); v.deiana83@gmail.com (V.D.); silvial_s@libero.it (S.L.d.S.S.); lau.puddu@tiscali.it (L.P.); francesca.fatteri@atssardegna.it (F.F.); aliceghiani@hotmail.it (A.G.); alicegergei@tiscali.it (A.L.); bcarpini@iol.it (B.C.); 2Department of Pharmacology, Dalhousie University, Halifax, NS B3H 4R2, Canada; 3Department of Mental Health and Addictions, ASL Cagliari, 09127 Cagliari, Italy; graziella.boi@aslcagliari.it (G.B.); annadele.pes@aslcagliari.it (A.P.)

**Keywords:** drunkorexia, food and alcohol disturbance, adolescents

## Abstract

Food and alcohol disturbance (FAD) is characterized by the association of alcohol use with compensatory behaviors such as restricting calories, physical activity and purging. Despite not being part of the current nosography, research has grown in the past 10 years, mostly on college students’ samples. In this study, we aim to describe the prevalence, characteristics and association of FAD with problem drinking (PD) and eating disorder risk (EDR) in a sample of Italian high school students. Participants were 900 high school students (53.6% males; mean age = 16.22) that were administered standardized questionnaires. Students who screened positive for PD, EDR and both were, respectively, 17.3%, 5.9% and 1.3%. Approximately one out four students reported FAD behaviors, mostly to control weight and by restricting calories, with higher prevalence and severity among those who screened positive for PD. Purging behaviors were rare overall (15.5%), but significantly more frequent in participants who screened positive for both PD and EDR (41.7%). FAD was more strongly associated with alcohol use severity than with ED symptom severity across all subgroups. FAD behaviors appear to be common in the Italian high school population and more strongly associated with PD. Future studies should investigate FAD’s impact on adolescents’ functioning and possible early interventions.

## 1. Introduction

Rates of alcohol use and disordered eating among adolescents and young adults have been well documented in the past 10 years [1,2] and represent a matter of concern in public health policy [3,4]. Italy has a lower rate of risky alcohol use among adolescents compared to other European countries, although with a high proportion of heavy episodic users [5]. Moreover, in a recent study conducted on a sample of high school students from Tuscany, a remarkable percentage of adolescents reported body image or eating concerns [6].

A particular pattern of behaviors that lie between problem drinking and disordered eating has been documented in the past decade, mostly among US college and university students [7,8]. It consists of associating alcohol use with compensatory behaviors commonly observed in the eating disorder population (calorie restriction, physical activity and purging behaviors such as self-induced vomiting and the use of laxatives or diuretics). By doing so, the individual can compensate for the calories ingested with alcohol and, when restricting, enhance its psychoactive effects (due to the quickest absorption rate on an empty stomach) [9,10]. The characterization of this phenomenon has been extended only recently to adult and adolescent non-US samples [11,12], although with inconsistent definitions. More generally, there is a lack of clear and standardized conceptualizations in the literature [9]. In order to promote the operationalization of the construct, Choquette et al. [13] have proposed the broad terminology “Food and Alcohol Disturbance” (FAD) to be used more consistently in the field. Moreover, other authors have recommended to discontinue the use of semantically inaccurate and pejorative terms frequently cited in prior literature, such as “drunkorexia” [14]. Therefore, we will use the term FAD in this research paper.

To this day, only few studies have explored FAD prevalence in Italy. Regarding young adults, Lupi et al. [15] reported a prevalence of 34.1% in their sample from the general population (age 18–26). This was in line with the results of Di Tata et al., [16] who reported a FAD prevalence of 38% in a comparable non-clinical sample. Regarding Italian adolescents, a series of studies published by the same research group reported a prevalence of FAD ranging from 12% to 34.6% [13,17,18] using different definitions. The prevalence, characterization and impact of FAD in high school populations is of great interest, as several studies have shown strong evidence-based effectiveness of school-based interventions for a large variety of adolescent risk behaviors, such as alcohol and substance use and unprotected sexual intercourse [19,20].

One of the most discussed matters regarding FAD pertains to its potential nosography. In fact, it remains unclear whether FAD is nosologically closer to alcohol use disorder (AUD) or to eating disorders (EDs). For instance, Hunt and Forbush [21] showed in their college students sample that disordered eating added slightly more incremental validity compared to alcohol use to the prediction of FAD in a regression model, especially in females. In another US college sample, FAD was associated with alcohol-related outcomes but not with bulimia [22]. Roosen and Mills [23] found in university students that participants with FAD had higher scores of disordered eating and alcohol use according to their motives (respectively, to control calories or to accelerate intoxication). In the study by Pompili and Laghi on Italian adolescents [13], both disordered eating and alcohol use contributed similarly to FAD in both genders. This gap of knowledge appears to be related to several reasons: (i) the small number of publications in the field; (ii) the inconsistency in FAD construct definitions and assessments across studies; (iii) differences in study design, psychometric instruments and statistical methodologies; (iv) the presence of potential confounders, such as possible undetected comorbidity with a clinical AUD or ED and different motives to engage in FAD behaviors (i.e., enhancing alcohol effect vs. weight control) [9]. Despite these limitations, some authors have proposed to classify FAD as an eating disorder [24].

In this study, we aimed to contribute to the body of research on FAD in adolescence and its relation with EDs and AUD. Secondary analysis was performed on data collected cross-sectionally on a non-clinical sample of Italian adolescents recruited for the validation study of the Italian version of the Compensatory Eating and Behaviors in Response to Alcohol Consumption Scale (CEBRACS) [25]. Participants were screened for problem drinking (PD) and eating disorder risk (EDR) and divided accordingly into the following four categories: (i) control (screened negative for both PD and EDR); (ii) PD (screened positive for PD but negative for EDR); (iii) EDR (screened positive for EDR but negative for PD); (iv) PD + EDR (screened positive for both PD and EDR).

Firstly, we described FAD prevalence in the different subgroups and segmented its characteristics in terms of compensatory behaviors’ timing (before, during or after alcohol use), type (restriction, physical activity and purging), reason (to enhance alcohol effects, weight control or both) and severity. To the best of our knowledge, this is the first study to thoroughly characterize FAD patterns in a sample of adolescents.

Secondly, we explored the relationship between FAD, EDs and AUD by (i) comparing participants with FAD to participants without FAD across the four subgroups in a range of demographic and psychopathological characteristics; (ii) comparing the control with FAD (Control+) to PD without FAD (PD−) and EDR without FAD (EDR−); (iii) exploring the association of FAD with AUD and ED symptom severity across subgroups.

We think that an accurate characterization of FAD among adolescents is important to implement its definition and to clarify its nosography and potential clinical relevance, as school represents a key setting to deliver targeted interventions for risky behaviors.

## 2. Materials and Methods

### 2.1. Participants

Study participants were recruited in a Cagliari (Italy) public high school. In Italy, public high schools are mixed gender and have a 5-year program, covering the education of students in a typical age span from 14 to 18 years old. Participants were administered a general socio-demographic questionnaire and the following questionnaires: CEBRACS [26]; Alcohol Use Disorders Identification Test (AUDIT) [27]; Eating Disorder Inventory-3 (EDI-3) [28]. Details of the assessment procedure have been reported in our primary analysis publication [25].

### 2.2. Measures

#### 2.2.1. Demographics

Participants reported gender, age, current tobacco use, and lifetime history of substance use.

#### 2.2.2. Food and Alcohol Disturbance (FAD)

FAD was investigated with the Compensatory Eating and Behaviors in Response to Alcohol Consumption Scale (CEBRACS). It is a 21-item self-administered questionnaire that measures the frequency of different types of FAD behaviors in the previous three months on a scale from 1 (never) to 5 (almost all the time) [26]. It is divided into three sections based on whether the FAD behavior happened before, during or after alcohol use. The items cover two reasons (weight control and to enhance alcohol effects) and several types (restricting, physical activity and purging) of FAD behaviors. Our research group has previously validated the Italian version of the CEBRACS [25] in the same sample used for this analysis, showing good internal consistency (Cronbach’s Alpha = 0.886) and concurrent validity using a 5-factor model. It should be noted that since the publishing of our work, research on other samples with confirmatory factor analysis was not able to replicate consistently any factor solution [29,30]. For the purpose of this paper, the presence of FAD was reported if participants scored at least 1 item ≥ 2. FAD severity was calculated as the total score derived from the sum of the individual item scores (range from 21 to 105). We defined three motives underlying FAD behaviors: (i) to enhance the alcohol effect (EAE-FAD); (ii) to control weight (WC-FAD); (iii) to both enhance the alcohol effect and control weight (EAEWC-FAD). We defined three timings of FAD in relation to alcohol use: (i) before alcohol use (FAD-Before); (ii) during alcohol use (FAD-During); (iii) after alcohol use (FAD-After). Finally, we defined three types of FAD compensatory behaviors: (i) restricting calories (FAD-Restricting); (ii) physical activity (FAD-Hyperactivity); (iii) purging (FAD-Purging). Details of the variables’ definitions are provided in the Appendix A.

#### 2.2.3. Eating Disorder Risk (EDR)

EDR was assessed with the Eating Disorder Inventory-3 (EDI-3) [28], specifically, the Italian version [31]. It is a validated self-administered questionnaire, consisting of 91 questions, widely used both in research and in clinical settings to assess ED populations. Single item scores are elaborated to obtain 12 index scores: 3 that measure ED symptoms (drive for thinness, DT; bulimia, B; body dissatisfaction, BD) and 6 that measure general psychological features commonly associated with EDs (low self-esteem, LSE; personal alienation, PA; interpersonal insecurity, II; interpersonal alienation, IA; interoceptive deficits, ID; emotional dysregulation, ED; perfectionism, P; asceticism, AS; maturity fear, MF). The first 3 index scores are combined to obtain an eating disorder risk composite score (EDRC). An EDRC percentile score exceeding 85° is indicative of a high clinical risk of EDs [28]. We used this threshold to define EDR participants. The raw score of EDRC was used to measure ED symptom severity. We also collected from the EDI-3 the information regarding lifetime history of self-induced vomiting after meals.

#### 2.2.4. Problem Drinking (PD)

PD was assessed with the Alcohol Use Disorders Identification Test (AUDIT), a comprehensive 10-question self-administered alcohol harm screening tool [27] developed by the World Health Organization (WHO). It investigates drinking habits in the previous 12 months. To the best of our knowledge, the AUDIT has not been validated in Italian adolescents. In a sample of 225 mixed-gender German high school students [32], a cut-off score of 6 was shown to have good sensitivity (0.79) and specificity (0.79) in identifying problem drinking (defined as current alcohol dependence, abuse or heavy episodic drinking according to a diagnostic standardized interview). Given the solid design of this validation study and the comparability of samples (European mixed-gender adolescents), we chose to apply this cut-off to define PD participants in our study. AUDIT total score was used as a measure of PD severity.

### 2.3. Statistical Analysis

For descriptive statistics, we used frequencies, mean and median values, as appropriate. A Chi-square test was used to compare FAD prevalence and characteristics between the four subgroups: (i) controls (screened negative for both PD and EDR); (ii) PD (screened positive for PD but negative for EDR); (iii) EDR (screened positive for EDR but negative for PD); (iv) PD + EDR (screened positive for both PD and EDR). To compare FAD severity among the four subgroups we used the Kruskal–Wallis H test with pairwise Mann–Whitney post-hoc tests with Bonferroni correction. To compare participants with FAD and participants without FAD across subgroups, we used a Chi-square test, Student’s t-test and Mann–Whitney U test, as appropriate. For between-group comparisons between Control+, PD− and EDR−, we used a Chi-square test, one-way ANOVA with Bonferroni post-hoc test and Kruskal–Wallis H test with pairwise Mann–Whitney post-hoc test, as appropriate. For these analyses, we set an alpha significance value of 0.0027, applying the Bonferroni correction for multiple comparisons in order to minimize type 1 errors. The normality of continuous variables’ distribution in compared subgroups of participants was ascertained by observing normality plots, skewness and kurtosis. Finally, we used multiple linear regression to investigate predictors of PD and ED symptom severity (dependent variables) in each subgroup, with gender (categorical), age (continuous), FAD (categorical), and lifetime substance use (categorical) as independent variables. The models were repeated, removing from the sample participants that reported engaging in FAD behaviors without weight control purposes in order to check the impact of narrowing the FAD phenotype. We used IBM SPSS Statistics 29.0.0.0 [33] for all analyses. An alpha significance value of 0.05 was utilized unless specified otherwise.

## 3. Results

The sample was composed of 965 high school students (55.0% males; mean age = 16.22 ± 1.66). Complete data were available for 900 participants (53.6% males; mean age = 16.22 ± 1.68). Participants excluded from analysis because of missing data (*n* = 65, 75.4% males) were significantly more likely to be males (χ^2^ = 11.673; *p* = 0.001).

### 3.1. Prevalence of PD, EDR and FAD

The majority of participants (66.9%; *n* = 646) scored at least one on the first question of the AUDIT questionnaire, thus reporting alcohol use in the previous 12 months. Students who screened positive for PD (but negative for EDR) were 167 (17.3%). Among them, 49.8% reported engagement in FAD behaviors in the three months prior to the assessment (*n* = 83).

The prevalence of EDR in the sample (excluding those positive to PD as well) was 5.9% (*n* = 57), with 16 participants endorsing FAD behaviors (28.1%).

Finally, 13 participants (1.3%) screened positive for both problem drinking and EDR. All but one (92.4%) reported engagement in FAD behaviors.

Among students who screened negative for both PD and EDR (68.7%; *n* = 663), 102 reported FAD (15.4%).

Overall, 23.7% of the whole sample reported engagement in FAD behaviors in the three months prior to the assessment (*n* = 213). The prevalence difference in FAD behaviors among groups was significant (χ^2^ = 122.343; *p* < 0.001, Table 1). Comparing the CEBRACS score among groups, we observed a significant different FAD severity (H = 137.410; *p* < 0.001), with post-hoc tests yielding the following result: PD + EDR > PD > EDR = CONTROL (with Bonferroni correction). The results are presented in Table 1.

### 3.2. FAD Motives and Patterns

Among those who engaged in FAD behaviors (*n* = 213), 42 participants (19.7%) did it to enhance the alcohol effect (EAE-FAD); 102 participants (47.9%) did it because of weight concerns (WC-FAD); and 69 participants (32.4%) did it for both reasons (EAEWC-FAD). Motives for FAD behaviors were significantly different among groups (χ^2^ = 25.785; *p* < 0.001, Table 2), with WC-FAD reported by the majority of participants in the control and EDR groups (respectively, 54.9% and 81.3%), while students in the PD and PD + EDR groups were more likely to engage in FAD for both reasons (EAEWC-FAD).

The most prevalent timing of FAD behavior was during alcohol consumption (70.4%; *n* = 150), without differences among groups. A significant difference was observed in FAD behaviors after alcohol use, which was more prevalent in the PD + EDR and PD groups compared to the control and EDR groups (Table 3).

Regarding compensatory behavior types, restricting calories was the most frequent in our sample (61.0%; *n* = 130). Purging behaviors were relatively rare in the overall sample (15.5%; *n* = 33) but significantly more frequent in the PD + EDR group (41.7%; *n* = 5; χ^2^ = 11.432; *p* = 0.010; Table 4).

### 3.3. Comparisons between Participants with and without FAD across Groups

Table 5 reports the results of the between-group comparisons between participants with FAD and participants without FAD across the control, PD and EDR groups. The analysis was not performed for the PD + EDR group, as only one participant did not report FAD. In the control group, participants with FAD were significantly more likely to report a lifetime history of substance use compared to those without (24.5% vs. 10.5%, χ^2^ = 14.035; *p* < 0.001). Moreover, they showed significantly higher scores in alcohol use severity (U = 46905.5; *p* < 0.001), drive for thinness (U = 35427.0; *p* < 0.001) and body dissatisfaction (U = 34761.0; *p* < 0.001). No other comparison was significant, considering an α = 0.0027. In the PD group, no significant difference between participants with and without FAD was detected. Two variables fell short of statistical significance: AUDIT score and emotional dysregulation, both higher in the FAD group, with a *p* = 0.004. ED symptom measures were comparable among the two groups. Finally, in the EDR group, participants with and without FAD differed only for AUDIT score, which was significantly higher in participants with FAD (U = 595.5; *p* < 0.001).

### 3.4. Comparisons between Participants in the Control+, PD− and EDR− Groups

Table 6 reports the results of the between-group comparisons between Control+, PD− and EDR−. All variables tested yielded significant differences, except for current cigarettes smoked (χ^2^ = 7.396; *p* = 0.025) and interpersonal insecurity (H = 10.635; *p* = 0.005). Regarding categorical variables, Control+ showed an intermediate phenotype, with 50% being males (vs. 4.9% in the EDR− group and 72.3% in the PD− group, χ^2^ = 48.8115; *p* < 0.001), 24.5% reporting a lifetime history of substance use (vs. 4.9% in the EDR− group and 59.5% in the PD− group, χ^2^ = 44.025; *p* < 0.001) and 10% reporting a lifetime history of self-induced vomiting after meals (vs. 43.9% in the EDR− group and 2.4% in the PD− group, χ^2^ = 42.837; *p* < 0.001). Regarding continuous variables, post-hoc test results were consistent with an intermediate phenotype for age, AUDIT score and drive for thinness. The Control+ group scored significantly higher on the AUDIT than the EDR− (but lower than the PD−) and significantly lower on the EDI Drive for Thinness than the EDR− (but higher than the PD−). Participants in the Control+ group were significantly older than those in the EDR− group, but significantly younger than those in the PD− group. For all other EDI scores (including the core scales for bulimia and body dissatisfaction), post-hoc tests showed that EDR− participants scored significantly higher, while the difference between Control+ and PD− was not significant (Table 6).

### 3.5. Linear Regression Analysis

Table 7 and Table 8 report the results of the linear regression analysis across the participant categories with PD severity (Table 7) and ED symptom severity (Table 8) as dependent variables. The model was not built for the PD + EDR group as the size was too small (*n* = 13) to support a multivariate model and only one participant did not report FAD.

FAD was significantly associated with PD severity across all of the three groups (Table 7), with a small effect size in the PD group (β = 0.242, *p* = 0.002), a moderate effect size in the control group (β = 0.361, *p* < 0.001) and a large effect size in the EDR group (β = 0.553, *p* < 0.001). In the PD group, it was the only significant independent predictor in the model, although the overall fit of the model was poor (explaining 5.1% of the variation in the outcome) and the standard errors around B coefficients were large. All three models had statistically significant F values. When narrowing the FAD phenotype by removing participants who did not engage in FAD behaviors for weight control purposes (EAE-FAD, *n* = 42), the results were comparable in terms of both significance and effect sizes (Appendix A).

Regarding ED symptom severity (Table 8), FAD was a significant predictor in the control and PD group, albeit with small β effect sizes (0.135 and 0.159, respectively). In the EDR group, no predictor was associated with the outcome, and the model was not significant (F = 1.058, *p* = 0.387). The predictor with the largest effect size was unsurprisingly gender, both in the control (β = −0.417, *p* < 0.001) and in the PD group (β = −0.508, *p* < 0.001), with males showing significant lower scores of ED symptom severity, accounting for the other variables in the model. The models in the control and PD groups were both significant (*p* < 0.001), but the overall fit was poor, with adjusted R^2^ values of 0.195 and 0.290, respectively (Table 8). Again, considering only participants with weight control concerns did not substantially modify significance and effect sizes of predictors (Appendix A).

## 4. Discussion

In this study, we aimed to describe FAD behavior prevalence and characteristics in a non-clinical sample of adolescents across different risk categories and to explore its relationship with EDs and AUD.

### 4.1. FAD Prevalence and Characteristics

Rates of participants who screened positive for PD, EDR and both were, respectively, 17.3%, 5.9% and 1.3%. Problem drinking rates in adolescence show great variation across countries [5]. Surveillance data from the Istituto Superiore di Sanità (Italian National Institute of Health) reported a rate of 21% of heavy drinkers among 15-year-olds in a national representative sample [34]. Regarding EDR, we found a lower prevalence compared to other adolescents samples [2,6]. However, most screening research is devoted to subclinical disordered eating [2] and uses assessment methods other than the EDI-3, which is not suited to screening purposes given the time required for administration and elaboration [35]. In contrast, we aimed to identify participants with a high risk of clinical EDs and chose a very conservative cut-off. Stachowitz et al. [36] used the EDRC score of the EDI-3 in a sample of US-only female adolescents and found that 10.77% of participants were at risk, which is in line with our findings given our mixed-gender sample (46.4% females).

FAD behaviors were reported by 23.7% of our sample. This prevalence is lower than that reported in studies on Italian young adults [15,16]. Laghi et al. [18] reported a prevalence of 34.6% in a sample of Italian high school students with the same FAD definition and assessment method that we used (CEBRACS score ≥ 22). Their number is higher compared to ours, but they included only participants who used alcohol. When limiting our sample to alcohol-using adolescents, the prevalence of FAD becomes comparable between the two samples (32.97%). Most participants in our sample who engaged in FAD behaviors reported doing it to control weight (47.9%) before or during alcohol use (respectively, 63.4% and 70.4%) and with restrictive behaviors (61.0%).

When comparing FAD prevalence between the subgroups, the difference was statistically significant, with FAD reported more frequently by participants who screened positive for PD. Importantly, participants who screened positive for both PD and EDR had a strikingly high rate of FAD (12 out of 13, 91.7%) and had the highest FAD severity among groups, reporting a median CEBRACS score of 28. Participants in the EDR group had a higher rate of FAD than controls (28.1% vs. 15.4%), but the severity score was significantly lower compared to both the PD and the PD + EDR groups and was not significantly different from the control group. When exploring reasons for FAD behaviors, we observed that in the control and EDR groups, most participants had weight control purposes, while in the PD and PD + EDR groups, participants engaged in FAD behaviors mostly to both control weight and enhance the alcohol effect. Across all subgroups, a low proportion of participants reported to engage in FAD behaviors only to enhance intoxication without weight control purposes (19.7% of all participants who reported FAD). Finally, both engaging in FAD behavior after alcohol consumption and purging FAD behaviors were more frequently reported in the PD and PD + EDR groups. Specifically, purging FAD behaviors were more prevalent in the PD + EDR group (41.7%) compared to the PD (20.5%), EDR (12.5%) and control (8.8%) groups. To our knowledge, no other study has characterized patterns of FAD behaviors among adolescents. It appears that FAD behavior motives, timing and type vary greatly across the investigated subgroups and that the presence of some characteristics (especially purging behaviors) could be suggestive of greater severity.

### 4.2. FAD, ED and AUD

In the control group, FAD was associated with a history of substance use and higher AUDIT scores but also with higher levels of drive for thinness and body dissatisfaction, in line with other studies [37]. Drive for thinness and body dissatisfaction include core cognitive dimensions of EDs, such as a fear of gaining weight, concerns with weight and dieting, and concerns with body size and shape [28]. These findings, coupled with the observation of the high prevalence and severity of FAD in the PD + EDR group, support the interplay of ED cognitions and alcohol use in adolescent FAD behaviors.

In the PD group, there was a trend for higher AUDIT scores and emotional dysregulation in participants with FAD, without significant differences in the core ED symptom scales (drive for thinness, bulimia and body dissatisfaction). Several studies on adolescents have found an association between FAD and emotional dysregulation [13,38,39]. This is not surprising, given that individuals with emotional dysregulation tend to have high rates of disordered eating, substance use and risky self-destructive behaviors [40]. It is important to note that the emotional dysregulation scale of the EDI-3 contains itself two items related to substance use and alcohol use [28].

In the EDR group, only the AUDIT score was significantly higher in participants with FAD, meaning that participants at high risk of clinical EDs who reported FAD did not differ from those without FAD for any psychological feature tested, except for severity of alcohol use.

When comparing the three clean-cut subgroups (Control+, EDR− and PD−), FAD appeared to be an intermediate phenotype for demographics, history of substance use, post-meal vomiting, severity of alcohol use and drive for thinness. However, for 10 out of 12 EDI index scores tested, no significant difference between the Control+ and the PD− group was detected, while scores in the EDR− group were consistently significantly higher. This was the case even for the core ED dimensions of bulimia and body dissatisfaction.

Although the lower sample size of the PD and EDR subgroups and the conservative alpha significance could have increased the risk of type 2 errors, these findings were consistent with the multiple linear regression model’s results. In fact, FAD predicted PD severity across all tested subgroups, while the association with ED symptoms was significant only in the control and PD groups and with smaller effect sizes. The largest effect sizes for predicting ED symptoms were for the female gender in the control and PD groups, while the model in the EDR group was not significant. Moreover, these results were insensitive to the removal of participants who engaged in FAD without weight control purposes.

According to these findings, we hypothesize that FAD could more closely relate to AUD than to EDs, even after narrowing the phenotype to weight control FAD, which is consistent with the observation that FAD frequency and severity were both higher in the PD group compared to the EDR group.

Our results are in line with those of Simons et al., [22] who used structural equation modeling in a sample of US college students and found the FAD, despite the association with cognitive features of EDs, namely, drive for thinness, was associated with alcohol-related outcomes but not with bulimia. However, other authors have found contrasting findings. Hunt and Forbush [21] used sequential linear regression to predict FAD severity in a sample of college students and found that ED symptom scores explained more variance in the outcome compared to the AUDIT score. Interestingly, to define FAD, they used a five-item assessment that included the following item: “I drank excessive amounts of alcohol so that I could vomit food I had eaten” [21]. This dimension is not included in the CEBRACS questionnaire that we used [25,26], where self-induced vomiting is explored only in relation to alcohol calories and not to food. Therefore, their FAD definition might explain the increased ED symptom effect sizes in predicting FAD. Moeck and Thomas [11] replicated Hunt and Forbush’s results in a sample of adult drinkers, but assessed FAD with the CEBRACS questionnaire. In their study, they used the Eating Pathology Symptoms Inventory [41] (EPSI) and the AUDIT to predict CEBRACS score with linear regression. The EPSI is a 45-item questionnaire and contains numerous items to investigate compensatory behaviors, without specifying whether in response to alcohol or food calories [41]. In our study, we used the EDRC EDI-3 score which measures ED cognitions and binge eating [28]. Thus, our different results could be explained not only by a different sample (adults vs. adolescents) but also by the different ED symptoms measured.

In our sample of adolescents stratified for risk of ED and PD, FAD was more strongly associated with alcohol use both in terms of prevalence and in terms of effect sizes. Notably, most of the literature on FAD consequences has examined alcohol-related problems, such as injuries, physical violence and unprotected sexual activity [42,43]. Considering this, we disagree with the proposal to classify FAD as an ED [25]. Even if cognitive ED symptoms and compensatory behaviors characterize FAD [7,8,9], our study suggests that FAD might be more likely to happen in the context of alcohol abuse. Thus, we think that FAD characterization and study might be better suited to the field of substance use disorders research. Moreover, in countries like Italy where ED and AUD services are separated, careful consideration is required to decide which service should be in charge of primary prevention and treatment. Even in terms of potential interventions, we speculate that it is unlikely that standard ED treatments could be effective without specifically targeting alcohol use. On the other hand, in the absence of comorbid EDs, some typical ED interventions might not be necessary (for instance, nutritional interventions).

### 4.3. Strengths, Limitations and Future Directions

The strengths of our study include the representative sample of high school students and the use of standardized validated instruments. We also acknowledge several limitations. Firstly, this is a secondary analysis. Therefore, this study was not originally designed to answer our research questions, and data to specifically clarify FAD nosography are lacking. Secondly, the analysis was explorative, increasing the risk of false-positive findings [44]. Thirdly, data were collected cross-sectionally and thus no causal relationship can be derived. Other limitations include the small size of the PD + EDR group and the limited generalizability of our results to the whole Italian adolescent population, given that school in Italy is mandatory only up to age 16 and given the variation in alcohol use and binge drinking rates across Italian regions [45].

Given the exploratory nature of our research, future studies should be specifically designed to directly test our hypothesis before drawing any conclusion on FAD nosography, possibly extending FAD investigation to clinical samples. Future research should also further characterize FAD in adolescence, especially with longitudinal studies. Moreover, studies should also address the potential consequences of FAD in adolescent populations. We think that it is of great importance to understand if FAD is associated with worse functioning or with later onset disorders such as EDs or AUD. In fact, the school setting could represent an opportunity to deliver early targeted interventions and reduce the burden of behavioral disorders in adolescents.

## 5. Conclusions

Even considering its inherent limitations, this study indicates that FAD behaviors appear to be common in the Italian high school population. Adolescents engaged in compensatory behaviors mostly to control weight and by restricting calories. Despite this, FAD was more strongly associated with problem drinking than with eating disorder risk in our sample. Future studies should investigate the impact of FAD on adolescents’ functioning and its longitudinal outcome. Research on FAD is still at the dawn, but it provides a window to early tailored interventions.

## Figures and Tables

**Table 1 ijerph-21-00083-t001:** Prevalence and severity of food and alcohol disturbance in the sample (*n* = 900). Participants are divided into categories based on results of problem drinking and eating disorder risk screening. FAD = food and alcohol disturbance. PD = Problem Drinking. EDR = eating disorder risk. M = median. IQR = interquartile range.

Group	Reporting FAD * *n*; %	FAD Severity ** M; IQR; Range
Control (*n* = 663)	102; 15.4%	21; 0; 21–42
PD (*n* = 167)	83; 49.7%	21; 5; 21–91
EDR (*n* = 57)	16; 28.1%	21; 1; 21–39
PD + EDR (*n* = 13)	12; 92.3%	28; 13.5; 21–42
Total Sample (*n* = 900)	213; 23.7%	21; 0; 21–91

* χ^2^ = 122.343; *p* < 0.001. ** H = 137.410; *p* < 0.001. (post-hoc: PD + EDR > PD > EDR = CONTROL).

**Table 2 ijerph-21-00083-t002:** Motives to engage in food and alcohol disturbance (FAD) behaviors across groups. PD = problem drinking. EDR = eating disorder risk. EAE = enhance alcohol effect. WC = weight control.

Group	EAE-FAD *n*; %	WC-FAD *n*; %	EAEWC-FAD *n*; %
Control (*n* = 102)	25; 24.5%	56; 54.9%	21; 20.6%
PD (*n* = 83)	15; 18.1%	28; 33.7%	40; 48.2%
EDR (*n* = 16)	1; 6.3%	13; 81.3%	2; 12.5%
PD + EDR (*n* = 12)	1; 8.3%	5; 41.7%	6; 50.0%
Total Sample (*n* = 213)	42; 19.7%	102; 47.9%	69; 32.4%

χ^2^ = 25.785; *p* < 0.001.

**Table 3 ijerph-21-00083-t003:** Timing of food and alcohol disturbance (FAD) in relation to alcohol consumption across groups. PD = problem drinking. EDR = eating disorder risk. AU = alcohol use.

Group	FAD-Before AU * *n*; %	FAD-During AU ** *n*; %	FAD-After AU *** *n*; %
Control (*n* = 102)	57; 55.9%	69; 67.6%	50; 49.0%
PD (*n* = 83)	59; 71.1%	61; 73.5%	50; 60.2%
EDR (*n* = 16)	12; 75%	11; 68.8%	7; 43.8%
PD + EDR (*n* = 12)	7; 58.3%	9; 75.0%	11; 91.7%
Total Sample (*n* = 213)	135; 63.4%	150; 70.4%	118; 55.4%

* χ^2^ = 5.656; *p* = 0.130. ** χ^2^ = 0.895; *p* = 0.827. *** χ^2^ = 9.734; *p* = 0.021.

**Table 4 ijerph-21-00083-t004:** Types of compensatory behaviors in food and alcohol disturbance (FAD) across groups. PD = problem drinking. EDR = eating disorder risk.

Group	FAD-Restrictive * *n*; %	FAD-Hyperactivity ** *n*; %	FAD-Purging *** *n*; %
Control (*n* = 102)	63; 61.8%	43; 42.2%	9; 8.8%
PD (*n* = 83)	45; 54.2%	42; 50.6%	17; 20.5%
EDR (*n* = 16)	13; 81.3%	9; 56.3%	2; 12.5%
PD + EDR (*n* = 12)	9; 75.0%	7; 58.3%	5; 41.7%
Total Sample (*n* = 213)	130; 61.0%	101; 47.4%	33; 15.5%

* χ^2^ = 5.378; *p* = 0.146. ** χ^2^ = 2.544; *p* = 0.467. *** χ^2^ = 11.432; *p* = 0.010.

**Table 5 ijerph-21-00083-t005:** Between-group comparisons (participants reporting food and alcohol disturbance, FAD+ vs. participants not reporting it, FAD−) in the control (*n* = 663), problem drinking (PD, *n* = 167) and eating disorders risk (EDR, *n* = 57) subgroups. Significant differences (*p* at significance level of α = 0.0027) are bolded.

	Control Group (*n* = 663)			PD Group (*n* = 167)			EDR Group (*n* = 57)		
Variable	FAD− (*n* = 561)	FAD+ (*n* = 102)	χ^2^/t/U; *p*	FAD− (*n* = 84)	FAD+ (*n* = 83)	χ^2^/t/U; *p*	FAD− (*n* = 41)	FAD+ (*n* = 16)	χ^2^/t/U; *p*
Male gender	*n* = 305; 54.4%	*n* = 51; 50.0%	χ^2^ = 0.662; *p* = 0.416	*n* = 60; 71.4%	*n* = 60; 72.3%	χ^2^ = 0.000; *p* = 1.000	*n* = 2; 4.9%	*n* = 1; 6.3%	χ^2^ = 0.000; *p* = 1.000
Age	# 15.98 ± 1.69	16.46 ± 1.55	t = 2.695; *p* = 0.004	# 17.15 ± 1.30	16.92 ± 1.47	t = −1.114; *p* = 0.133	# 15.34 ± 1.59	16.44 ± 1.59	t = −2.338; *p* = 0.013
Current Cigarettes Smoked	*n* = 119; 21.2%	*n* = 36; 35.3%	χ^2^ = 8.785; *p* = 0.003	*n* = 42; 50.0%	*n* = 50; 60.2%	χ^2^ = 1.380; *p* = 0.240	*n* = 11; 26.8%	*n* = 6; 37.5%	χ^2^ = 0.220; *p* = 0.639
History of Substance Use	***n* = 59; 10.5%**	***n* = 25; 24.5%**	**χ^2^ = 14.035; *p* < 0.001**	*n* = 50; 59.5%	*n* = 46; 55.4%	χ^2^ = 0.144; *p* = 0.704	*n* = 2; 4.9%	*n* = 4; 25.0%	χ^2^ = 3.042; *p* = 0.081
History of Self-Induced Vomiting	*n* = 22; 3.9%	*n* = 10; 10.0%	χ^2^ = 5.504; *p* < 0.019	*n* = 2; 2.4%	*n* = 8; 9.9%	χ^2^ = 2.859; *p* = 0.091	*n* = 18; 43.9%	*n* = 8; 50.0%	χ^2^ = 0.014; *p* = 0.905
AUDIT score	*** 1 ± 2**	**3 ± 3**	**U = 46,905.5; *p* < 0.001**	* 8 ± 4	9 ± 6	U = 4375.0; *p* = 0.004	*** 0 ± 1**	**2.5 ± 4**	**U = 595.5; *p* < 0.001**
EDI Drive for Thinness	*** 2 ± 6**	**5 ± 11**	**U = 35,427.0; *p* < 0.001**	* 1 ± 5	4 ± 11	U = 4116.5; *p* = 0.040	# 23.46 ± 3.52	23.88 ± 4.76	t = 0.359.0; *p* = 0.361
EDI Bulimia	* 2 ± 5	3 ± 6	U = 32,358.0; *p* = 0.033	* 4 ± 6	5 ± 7	U = 3912.5; *p* = 0.171	# 14.10 ± 6.07	12.38 ± 6.47	t = −0.919; *p* = 0.183
EDI Body Dissatisfaction	*** 8 ± 12**	**11 ± 14**	**U = 34,761.0; *p* < 0.001**	* 8.5 ± 11	9 ± 12	U = 3641.5; *p* = 0.618	# 33.54 ± 5.06	30.63 ± 4.38	t = −2.024; *p* = 0.024
EDI Low Self-Esteem	* 3 ± 8	5 ± 8	U = 31,327.5; *p* = 0.123	* 4.5 ± 8	4 ± 10	U = 3669.0; *p* = 0.556	# 14.17 ± 6.48	10.75 ± 6.95	t = −1.754; *p* = 0.042
EDI Personal Alienation	* 4 ± 6	5 ± 7	U = 30,517.5; *p* = 0.282	* 5 ± 7	7 ± 7	U = 3966.0; *p* = 0.124	# 13.83 ± 6.64	12.0 ± 6.21	t = −0.951; *p* = 0.173
EDI Interpersonal Insecurity	* 6 ± 8	6 ± 6	U = 28,435.0; *p* = 0.921	* 7 ± 7	6 ± 6	U = 3119.0; *p* = 0.239	# 10.41 ± 5.50	9.19 ± 5.55	t = −0.755; *p* = 0.227
EDI Interpersonal Alienation	* 6 ± 6	7 ± 6	U = 33,652.0; *p* = 0.004	* 7 ± 7	7 ± 6	U = 3406.5; *p* = 0.799	# 11.44 ± 5.62	9.06 ± 4.95	t = −1.481; *p* = 0.072
EDI Interoceptive Deficits	* 5 ± 9	8 ± 10	U = 33,512.5; *p* = 0.006	* 7 ± 10	11 ± 11	U = 4286.5; *p* = 0.010	# 17.24 ± 8.51	13.13 ± 9.88	t = −1.569; *p* = 0.061
EDI Emotional Dysregulation	* 4 ± 7	5 ± 7	U = 32,924.5; *p* = 0.015	* 5 ± 8	10 ± 9	U = 4384.5; *p* = 0.004	# 12.76 ± 5.44	10.56 ± 6.08	t = −1.323; *p* = 0.096
EDI Perfectionism	* 6 ± 7	7 ± 6	U = 30,993.5; *p* = 0.179	* 7 ± 7	9 ± 10	U = 4048.0; *p* = 0.071	# 11.22 ± 5.91	11.56 ± 7.04	t = 0.186; *p* = 0.426
EDI Asceticism	* 5 ± 6	5 ± 6	U = 32,253.0; *p* = 0.040	* 5 ± 5	7 ± 6	U = 4213.5; *p* = 0.019	# 13.12 ± 3.95	12.94 ± 5.69	t = −0.019; *p* = 0.453
EDI Maturity Fear	* 10 ± 7	10 ± 6	U = 28,024.0; *p* = 0.741	* 10 ± 7	9 ± 7	U = 3504.5; *p* = 0.953	# 16.78 ± 5.82	16.81 ± 7.72	t = 0.017; *p* = 0.493

* = median ± interquartile range, Mann–Whitney U; # = mean ± standard deviation, Student’s t.

**Table 6 ijerph-21-00083-t006:** Between-group comparisons. Significant differences (*p* at significance level of α = 0.0027) are bolded. EDR− = eating disorder risk without food and alcohol disturbance. Control+ = control with food and alcohol disturbance. PD− = problem drinking without food and alcohol disturbance.

Variable	EDR− (*n* = 41)	Control+ (*n* = 102)	PD− (*n* = 84)	χ^2^/*F*/*H*	Post-Hoc Tests
**Male Gender**	*n* = 2; 4.9%	*n* = 51; 50.0%	*n* = 60; 72.3%	χ^2^ = 48.8115; *p* < 0.001	/
**Age**	# 15.34 ± 1.59	16.46 ± 1.55	17.15 ± 1.30	*F* = 21.024; *p* < 0.001	EDR− < Control+ < PD−
Current Cigarettes Smoked	*n* = 11; 26.8%	*n* = 36; 35.3%	*n* = 42; 50.0%	χ^2^ = 7.396; *p* = 0.025	/
**History of Substance Use**	*n* = 2; 4.9%	*n* = 25; 24.5%	*n* = 50; 59.5%	χ^2^ = 44.025; *p* < 0.001	/
**History of Self Induced Vomiting After Meals**	*n* = 18; 43.9%	*n* = 10; 10.0%	*n* = 2; 2.4%	χ^2^ = 42.837; *p* < 0.001	/
**AUDIT Score**	* 0 ± 1	3 ± 3	8 ± 4	*H* = 182.980; *p* < 0.001	EDR− < Control+ < PD−
**EDI Drive for Thinness**	* 24 ± 5	5 ± 11	1 ± 5	*H* = 104.216; *p* < 0.001	EDR− > Control+ > PD−
**EDI Bulimia**	* 13 ± 10	3 ± 6	4 ± 6	*H* = 65.544; *p* < 0.001	EDR− > Control+ = PD−
**EDI Body Dissatisfaction**	* 35 ± 7	11 ± 14	8.5 ± 11	*H* = 92.263; *p* < 0.001	EDR− > Control+ = PD−
**EDI Low Self-Esteem**	* 14 ± 8	5 ± 8	4.5 ± 8	*H* = 47.287; *p* < 0.001	EDR− > Control+ = PD−
**EDI Personal Alienation**	* 14 ± 9	5 ± 7	5 ± 7	*H* = 41.833; *p* < 0.001	EDR− > Control+ = PD−
EDI Interpersonal Insecurity	* 11 ± 8	6 ± 6	7 ± 7	*H* = 10.635; *p* = 0.005	/
**EDI Interpersonal Alienation**	* 11 ± 8	7 ± 6	7 ± 7	*H* = 17.905; *p* < 0.001	EDR− > Control+ = PD−
**EDI Interoceptive Deficits**	* 17 ± 14	8 ± 10	7 ± 10	*H* = 31.596; *p* < 0.001	EDR− > Control+ = PD−
**EDI Emotional Dysregulation**	* 13 ± 8	5 ± 7	5 ± 8	*H* = 32.593; *p* < 0.001	EDR− > Control+ = PD−
**EDI Perfectionism**	* 12 ± 10	7 ± 6	7 ± 7	*H* = 14.776; *p* < 0.001	EDR− > Control+ = PD−
**EDI Asceticism**	* 13 ± 6	5 ± 6	5 ± 5	*H* = 66.801; *p* < 0.001	EDR− > Control+ = PD−
**EDI Maturity Fear**	* 17 ± 9	10 ± 6	10 ± 7	*H* = 39.761; *p* < 0.001	EDR− > Control+ = PD−

# = mean ± standard deviation, one-way ANOVA; * = median ± interquartile range, Kruskal–Wallis H test.

**Table 7 ijerph-21-00083-t007:** Results of linear regression analysis with food and alcohol disturbance (FAD, 0 = not reported; 1 = reported), gender (0 = females, 1 = males), age (continuous) and lifetime substance use (SU; 0 = not reported, 1 = reported) as predictors and problem drinking (PD) severity as dependent variable. Results are reported for all categories. Significant predictors (α = 0.05) are bolded. PD = problem drinking; EDR = eating disorder risk.

Group	Dependent Variable: PD Severity				
	Variables	B (95% CI); SE	β	*p*	Adjusted R^2^; F; *p*
Control (*n* = 663)	(Constant)	−3.519 (−4.521–−2.517); 0.510		<0.001	0.329; 82.038; <0.001
	**Gender**	0.627 (0.062–0.472); 0.105	0.082	0.011	
	**Age**	0.278 (0.216–0.340); 0.032	0.286	<0.001	
	**FAD**	1.628 (1.342–1.915); 0.146	0.361	<0.001	
	**Lifetime SU**	1.083 (0.769–1.398); 0.160	0.221	<0.001	
PD (*n* = 167)	(Constant)	4.548 (−4.996–14.093); 4.833		0.348	0.051; 3.213; 0.014
	Gender	1.273 (−0.448–2.994); 0.871	0.112	0.146	
	Age	0.205 (−0.354–0.763); 0.283	0.056	0.470	
	**FAD**	2.468 (0.939–3.998); 0.775	0.242	0.002	
	Lifetime SU	−0.064 (−1.636–1.507); 0.796	−0.006	0.936	
EDR (*n* = 57)	(Constant)	−3.667 (−6.928–−0.406);1.625		0.028	0.448; 12.378; <0.001
	Gender	−0.417 (−1.899–1.065); 0.739	−0.056	0.575	
	**Age**	0.281 (0.070–0.492); 0.105	0.279	0.010	
	**FAD**	2.034 (1.231–2.838); 0.400	0.553	<0.001	
	Lifetime SU	0.145 (−0.984–1.273); 0.562	0.027	0.798	

**Table 8 ijerph-21-00083-t008:** Results of linear regression analysis with food and alcohol disturbance (FAD, 0 = not reported; 1 = reported), gender (0 = females, 1 = males), age (continuous) and lifetime substance use (SU; 0 = not reported, 1 = reported) as predictors and eating disorder (ED) symptom severity as dependent variable. Results are reported for all categories. Significant predictors (α = 0.05) are bolded. PD = problem drinking; EDR = eating disorder risk.

Group	Dependent Variable: ED Symptom Severity				
	Variables	B (95% CI); SE	β	*p*	Adjusted R^2^; F; *p*
Control (*n* = 663)	(Constant)	37.588 (27.329–47.848); 5.225		<0.001	0.195; 41.183; <0.001
	**Gender**	−12.731 (−14.834–−10.629); 1.071	−0.417	<0.001	
	**Age**	−0.841 (−1.474–−0.207); 0.323	−0.092	0.009	
	**FAD**	5.698 (2.763–8.633); 1.495	0.135	<0.001	
	**Lifetime SU**	3.709 (0.491–6.928); 1.639	0.081	0.024	
PD (*n* = 167)	(Constant)	34.846 (10.880–58.813); 12.137		0.005	0.290; 17.944; <0.001
	**Gender**	−16.720 (−21.041–−12.399); 2.188	−0.508	<0.001	
	Age	−0.150 (−1.551–1.252); 0.710	−0.014	0.833	
	**FAD**	4.710 (0.869–8.550); 1.945	0.159	0.017	
	Lifetime SU	−2.602 (−6.549–1.345); 1.999	−0.087	0.195	
EDR (*n* = 57)	(Constant)	74.325 (55.710–−92.940); 9.277		<0.001	0.004; 1.058; 0.387
	Gender	−0.003 (−8.465–8.458); 4.217	0.000	0.999	
	Age	−0.211 (−1.415–0.993); 0.600	−0.049	0.726	
	FAD	−4.041 (−8.627–0.546); 2.286	−0.258	0.083	
	Lifetime SU	0.247 (−6.194–6.688); 3.210	0.011	0.939	

## Data Availability

Data are available on request from the authors.

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
