# Peer review of "Food and Alcohol Disturbance in High School Adolescents: Prevalence, Characteristics and Association with Problem Drinking and Eating Disorders"

_ijerph, 2024, doi:10.3390/ijerph21010083_

Round 1

Reviewer 1 Report

Comments and Suggestions for Authors

The manuscript reports the findings of the prevalence and characteristics of food and alcohol disturbance (FAD) among a large sample of Italian adolescents who participated in the study through their high school. The study is overall strong and utilized a large sample of adolescents from a public high school, used a comprehensive assessment of FAD needed to comprehensively classify and characterize types of FAD behaviors (e.g., timing relative to drinking occasions and motives for FAD engagement). The study revealed differences in patterns of FAD across adolescents engaging in problem drinking and with elevated eating disorder risk, as well as differences in motives for engagement in FAD behaviors. I have several comments and suggestions, detailed below.

1. I appreciate that the authors use the term food and alcohol disturbance to examine the phenomenon, and that they mention in the introduction they do not use the term "drunkorexia" (despite its prior use in the literature) because it is both scientifically inaccurate and pejorative. However, the authors do still devote an entire paragraph at the beginning of the introduction section to describing the term drunkorexia and the origin of its use, before making the case that they will not use it throughout the document. I recommend removing the focus and background description on the drunkorexia term, and instead focus on describing the prior literature on food and alcohol disturbance, defined broadly. (At most, the authors could briefly mention that in the prior literature, a term that was used to describe the phenomenon with scientifically inaccurate and pejorative.) 

2. Was the sample of high school adolescents representative of students in the school? If so, this is a strength of the sample that should be stated directly. 

3. For the AUDIT cut off score of 6, the authors use one citation of a German adolescent high school sample as the justification. However, the AUDIT is used globally and there is an extensive body of literature addressing AUDIT scores across countries and ages. Thus, the justification doesn't seem to fully consider the available literature. This cut point may be consistent with the broader literature, but that should be more clearly specified (and if it's not, it should be more strongly justified).

4. In the introduction and discussion, the authors mention the question of whether FAD should be considered as an eating disorder or substance use disorder, and discuss this concept informed by their data. I think this concepts steps beyond what the authors can say with their data, and that such a study would require more intensive methodology to focus specifically on the alignment of FAD with other types of disorders, probably considered more broadly (e.g., HiTOP model). Furthermore, I don't really think that classifying FAD as one or the other is scientifically very useful, as per its definition, FAD contains elements of both eating and alcohol disturbances. 

Reviewer 2 Report

Comments and Suggestions for Authors

Food and Alcohol Disturbance in High School Adolescents Prevalence, Characteristics, and Associations with Problem Drinking and Eating Disorders

Summary: Thank you for the opportunity to review. The authors provide an in-depth examination of the prevalence rates of FAD in Italian adolescents. This is one of the only papers to have examined FAD as robustly and in a sample of adolescents in a country other than the US. The paper also attempts to determine whether FAD is more closely related to AUD or ED behaviors. While the paper has some minor addressable issues, it holds value to researchers examining this phenomenon and provides data contributing to a better understanding of FAD.

Introduction:

I understand the use of acronyms for groups/ FAD phenotypes; however, this often makes the paper difficult to follow. Please use fewer acronyms for clarity. (e.g., FAD-B-AU could simply be FAD Before).

Should cite the original paper proposing the terminology FAD, most appropriately would be on page 2 line 47-50

Citation on line 79 should be fore Thompson-Memmer “Despite these limitations, some authors have proposed to classify FAD as an Eating 78 Disorder” not Pinna et al., Please check all citations for accuracy

Methods/ Results

Section 3.1: Is the first question of the AUDIT your alcohol use prevalence in the sample? It would be helpful to make this clear for those not familiar with the AUDIT.

I am confused by the rationale for the linear regressions. The authors have extensively examined the prevalence rates by time, behavior, motive within different groups of adolescents and yet in the linear regressions entered one FAD “severity” score. The meaning and utility of the total score on the CEBRACS has been questioned in the literature. While it may provide a severity score, I think that examining motive for engagement may have more utility in understanding if FAD is related to problem drinking and eating disorder risk differentially. The use of a total score doesn’t seem to follow the logic to examine the prevalence in such exhaustive groups being that these behaviors may be different/differentially predict outcomes.

Further, the regressions are repeated for those who only did not engage in weight control purposes (i.e., those who only engaged in FAD for only enhancement of alcohol); however, there are still a large number of participants who engaged in FAD behaviors for both motives (n = 69) meaning they engage in FAD for alcohol motives as well. The rationale for this was weak and it didn’t add much to the content of the current paper. If the authors are interested in examining the contributions of narrowing FAD phenotype they should examine regressions with only the EAEWC-FAD group to make the interpretation more clear.

What was the rationale for the specified p = 0.001? There are 18 comparisons for each group differences. Bonferroni-Holm or Benjamini-Hochberg would be more lenient. For example, with Bonferroni-Holm and Benjamini-Hochberg corrections the adjusted p-value for your most significant result would be p = 0.0027. While considering type I error is important so is type II error especially given the exploratory nature of the current paper.

Table 4: I think it would be informative to have the full range of the FAD severity scores in addition to IQR.

The tables 5-7 are difficult to read. It would be useful to create some kind of visual breaks and group variables by how the statistics are reported. For example, tables could be organized as all variables needing to be reported as n;% then mean and standard deviation, then median and IQR. You can then break the table up using these as subheadings. It would be helpful to have the full row bolded for variables that are significant. 

Comments on the Quality of English Language

There are just a few minor editing issues.

Reviewer 3 Report

Comments and Suggestions for Authors

Dear Authors,

the following comments are intended to strengthen the manuscript:

Abstract: The information about the place (country) where the study was conducted is mention in the end of this part. I would suggest to moved it to the methodology part (more in the beginning). In the case of eating behaviors, eating disorders and alcohol consumption, socio-cultural conditions are of great importance.

Keywords: It is rather uncommon to use only the abbreviation (CEBRACS). It is worth indicating the age group that the study concerns as a keyword.

Introduction: This part is well researched. The only thing I'm missing is an opinion justifying conducting research in this specific age group?

Materials and methods: Can the selected region be considered representative of Italy? What were the reasons for choosing Cagliari? Although the authors refer to the previous publication in the study description, I believe that for the reader's convenience, this information could be added in the form of an annex. I lack information about the consent of the ethics committee (especially since the study involved young people under 18 years of age).

Results: In table 1 in my opinion it is enough to present % of participants reporting FAD (numbers are sum up to 100% and there are only two option: reporting and not reporting. I cannot see marked values with significant differences? PD+EDR is a very small group of only 12 subjects. It should be mention in limitation of the study. Table 2 – significant differences are not marked? The authors could consider presenting table 5 in the beginning of the results section. Tables 5,6,7 are difficult to follow as they are presented on separate pages (headers are missing for the second page) – the Authors should consider re-formatting them.

Discussion: I really appreciate this part: it is well-constructed and logical. However, it is also quite long and it would be easier for the reader to navigate through the text if it were divided into sections.

Round 2

Reviewer 1 Report

Comments and Suggestions for Authors

The authors addressed all of my prior comments. I have no further suggestions before publication.